# Examining the Sustainability of Contributions of Competing Core Organizational Capabilities in Response to Systemic Economic Crises

Ali O. Jifri [1,*], Paul Drnevich [2], William Jackson [2] and Ron Dulek [2]

1   Department of Public Administration, Faculty Economics and Business Administration,
    King Abdulaziz University, Jeddah 21589, Saudi Arabia
2   Department of Management, College of Business, University of Alabama, Tuscaloosa, AL 35487, USA
*   Correspondence: aojifri@kau.edu.sa; Tel.: +966-504-642-851

**Abstract:** A dynamic capability view is used in this study to explain how organizational capabilities operate effectively and efficiently in stable environments and respond dynamically to changing conditions in their operating environments. Such capabilities enable organizations to both create and sustain their performance. When faced with a systemic change in the environment, such as a global economic crisis, organizational capabilities may no longer contribute effectively to sustain organizational performance or their survival. In this study, we examine the effectiveness and sustainability of organizational capabilities in response to a systemic economic crisis. We do so through examining these issues in a broad multiyear sample of U.S. credit unions through the global financial crisis. In this context, organizations utilized two types of competing capabilities: explorative capabilities to increase revenues and/or exploitative capabilities to reduce expenses. The effectiveness of these capabilities and the sustainability of the resulting performance implications of their combined deployment remains under-theorized and insufficiently examined, particularly under conditions of high economic uncertainty. We examine these issues using a sample of 1127 large U.S. credit unions collecting comparative data during a period of economic stability from 2001 to 2004 and during a period of economic instability from 2006 to 2009, before and after the 2008 global financial crisis. We perform multiple regression analysis to examine the contributions and sustainability of organizational capabilities to relative performance. Interestingly, we find that in stable times, the explorative capability to increase revenues appears to contribute more to performance, while in the crisis period, the exploitative capability to reduce expenses appears to contribute more to performance. Further, the combined effect of deploying both "competing" capabilities simultaneously is related to performance only when the environment is stable and can be detrimental during a crisis. The results suggest that using expense decreasing capabilities (but not revenue increasing capabilities or both combined) is better when facing an economic crisis.

**Keywords:** dynamic capabilities; crisis; capabilities; ambidexterity; sustained performance; uncertainty

## 1. Introduction

Organizations operate in dynamic external environments which may experience significant changes in operating conditions over time. Economic conditions in particular, may quickly change, especially during periods of economic crisis, pandemics, wars, and/or natural disaster. As a result of recent and ongoing crises, such as the COVID-19 pandemic, climate change, regional wars, global economic uncertainty has increased, and organizations will likely face even greater uncertainty in the future.

Contingency theory argues in part that effective decisions for the deployment of capabilities are dependent on the external environment [1]. In the strategic management field, a growing body of research advocates using both operational and dynamic organizational capabilities to help firms better align themselves with their external environment, [1–4].

Operational capabilities are the day-to-day operating routines and processes of an organization [4]. Dynamic capabilities change the resource and capability base of these routines and processes to match changes in the external environment [4,5]. These capabilities allow firms to sustain performance even in unpredictable and changing environments. Dynamic capabilities can lead to better performance when the environment is changing through the resources that firms manage, change, and reconfigure [2]. The literature offers fairly strong support for this contribution, and also notes different types of dynamic capabilities, two of which, exploitative capabilities and explorative capabilities, we examine here [6–8]. These two capabilities are dynamic in that they change resources and ordinary capabilities to achieve better fit with the environment to improve performance.

Research on dynamic capabilities offers fairly strong support for the positive relationship between dynamic capabilities and performance [9]. However, support for its sustainability is more limited, and more research is needed to determine whether firms can sustain these performance benefits over time. Furthermore, dynamic capabilities are context-dependent, meaning that there are unique types of capabilities and the value they provide may differ in different contexts [9]. In the banking and credit union context, such performance contributions involve using dynamic capabilities to change (e.g., improve declining performance in a crisis) by either enhancing revenues or reducing expenses. We examine the effectiveness of such explorative capabilities to increase revenues through market expansion (i.e., diversification, M&As, and alliances) and exploitative capabilities to reduce costs through efficiency.

Moreover, the combined 'ambidextrous' (or 'competing') contributions of these mechanisms, which we also theorize and examine, remain under-theorized and largely unexamined, especially in uncertain environments. The effects of simultaneously using both explorative and exploitive capabilities is unclear. We also lack a clear understanding of the implications of high levels of environmental disturbance for the performance contributions of these capabilities. We know little about which capabilities or capability combinations are more effective for crisis response. To address these gaps, we examine the performance contributions of these two classes of 'core' capabilities: (1) the explorative capability to enhance revenues and (2) the exploitive capability to reduce cost or expenses. The answers to these questions will help managers understand how to better sustain performance through different economic conditions.

In this study, we seek to evaluate and validate the direct contributions of explorative and exploitative capabilities to relative performance, i.e., how the two types of capabilities perform in the same context over time. Second, we investigate whether the interaction effects of the capabilities are synergistic or sub-additive. Most past studies have examined dynamic capabilities in either a stable environment or in a dynamic environment. Few studies have examined them during both stable and crisis periods (see [10] for one exception). We do so using a sample of 1117 large U.S. credit unions (with assets of over $100 million) in two periods, from 2001 to 2004, a relatively stable period, and from 2006 to 2009 during the 2008 global financial crisis. To assess the sustainability of performance, we used longitudinal data from these two periods. Few studies have used such an approach to compare periods of economic stability and instability to test the contributions of capabilities.

## 2. Theory Development

According to configuration theory, numerous internal and external factors align together to find a suitable fit. As a result, organizational capabilities change to match contingencies in the external environment [1]. Therefore, the effectiveness of organizational capabilities depends on their alignment with factors in the external environment [11]. Contingencies may change the value of dynamic capabilities. Consequently, adjusting organizational capabilities could help in achieving sustained performance.

The strategic management literature provides several theories about achieving effectiveness and efficiency. Transaction cost economics is concerned with achieving efficiency by reducing the costs associated with producing a product or performing a service [12–14].

Similarly, one of Michael Porter's commonly taught 'generic strategies' suggests firms can earn above average profits by being more efficient through reducing costs below competitors [15]. Other theories, such as configuration theory and resource dependence theory, are concerned with effectiveness. Effectiveness is the ability to reach organizational goals, such as creating value that consumers are willing to pay for, which increases revenue. These two approaches offer direct contributions to firm performance through either increasing revenues or decreasing costs. Within an organization, resource limitations may prevent them from simultaneously being both effective and efficient, or even achieving an optimal balance between the two (i.e., being ambidextrous) [16].

For the context of our study, we define explorative capabilities as the organization's ability to effectively manage and enhance its revenue flows relative to its asset base (i.e., the ratio of operating revenues to assets). Conversely, we define exploitative capabilities as the organization's ability to efficiently manage and reduce its expenses relative to its asset base. Consistent with best practices, we focus on change in performance relative to competitors [17,18]. We use the dynamic capability literature to help us predict what capabilities will lead to better performance in different environmental conditions. We define the general concept of dynamic capabilities consistent with the foundational literature (e.g., [2,5]) as firm-level routines or processes for acquiring, integrating, reconfiguring, and extending or modifying resources [2,4,19,20].

Liu, Yu and Wu [7] (2019) suggest explorative capabilities include activities such as innovation, risk taking, flexibility and experimentation, while exploitative capabilities improve operational capabilities and make them more efficient [7]. The context in which these capabilities operate is considered an important dimension in dynamic capabilities research. Fainshmidt et al. [9] (2016) stressed that the economic context could have an important impact on the value contribution of capabilities. What is valuable in one context may not be valuable in another context. Therefore, we study both explorative and exploitative capabilities in different economic contexts. In the following sections we discuss the relative performance implications of explorative and exploitative capabilities individually and in combination (ambidexterity).

## 2.1. General Contributions of Explorative Capabilities to Sustained Performance

Explorative capabilities include growth strategies such as seeking new customers and markets (e.g., engaging in M&As and alliances), which leads to higher revenues, and therefore, should theoretically contribute to performance (e.g., [2,21,22]). Research has shown that explorative dynamic capabilities lead to positive outcomes (i.e., innovation) and improved performance [23,24]. We expect that, in times of high uncertainty, economic stagnation, or contraction, an average firm's revenues will likely decrease or contract. Conversely, in times of economic growth, an average firm's revenues should increase. For example, one of the ways financial institutions can increase revenue is by increasing fees and loans to both current and new customers. It is difficult for customers and businesses with fewer resources (i.e., small and young businesses) to obtain loans from large traditional commercial lending institutions during an economic downturn. Some lending institutions, therefore, enter such a market in an effort to increase lending activities and enhance their revenues. Such value creation and related cash inflows should be directly and positively correlated with firm performance.

In a crisis situation that results in difficult economic conditions, consumers' buying power (and firm revenues) may be reduced. Therefore, efforts to reach new customers or improve revenues from existing customers may fail. However, explorative capabilities to provide new products (and/or reach new consumers) collectively should improve performance by increasing revenues relative to its asset base. Firms with higher levels of explorative capabilities should experience higher performance than firms with lower levels. Therefore, the explorative capabilities needed to effectively manage a firm's revenue ratio should be positively associated with relative performance. Generating revenue by selling products or services is a prerequisite to generating a return on assets or investments.

Therefore, if a firm can generate revenue, it is expected to make more profit and perform better. This leads to the following hypotheses.

**Hypothesis 1a.** *In stable periods, the explorative capability to effectively manage a firm's revenues is positively associated with relative performance, such that organizations that can increase revenues are more likely to experience higher relative performance than organizations that cannot.*

**Hypothesis 1b.** *In crisis periods, the explorative capability to effectively manage a firm's revenues is positively associated with relative performance, such that organizations that can increase revenues are more likely to experience higher relative performance than organizations that cannot.*

### 2.2. General Contributions of Exploitative Capabilities to Sustained Performance

Similarly, exploitative capabilities to efficiently manage a firm's expenses should be directly and positively associated with its performance. Since there are costs associated with selling products or services (or generating a return on a loan or investment), reducing those costs should have a positive association with a firm's return, and in turn, firm performance. Such efficiency capabilities reduce costs and, through such contributions, should theoretically contribute to firm performance (e.g., [12–14]).

In times of economic growth, expenses should remain stable or increase. For example, during such times, firms are more confident about seeking expansion and are less concerned with expenses. Additional opportunities for expansion are likely to result in increased expenses. In times of high uncertainty, economic stagnation, or contraction, firms often stop or delay expansion plans such as M&As and vertical integration [25]. The quickest way to improve performance is to reduce expenses. It is the most operationally efficient way for a firm to improve its performance. Consequently, some firms may reduce their labor force (downsize) in order to reduce costs. Others may choose to reduce advertising expenses, producing an immediate reduction in expenses and a positive effect on the bottom line. For example, trimming an advertising budget from $600,000 to $350,000 a year instantly contributes an additional $250,000 to the bottom line. Firm performance should be directly and positively related to cost reduction and related decreases in cash outflows. Thus, a firm's ability to efficiently manage expenses will be positively associated with its relative performance, and firms with higher levels of these efficiency capabilities should experience higher performance relative to those with lower levels of these capabilities. Therefore, exploitative capabilities to efficiently manage a firm's expenses should be positively associated with relative performance. This line of argumentation suggests the following two hypotheses.

**Hypothesis 2a.** *In stable periods, the exploitative capability to efficiently manage a firm's expenses is positively associated with relative performance, such that organizations that can lower expense ratios are more likely to experience higher relative performance than organizations that cannot.*

**Hypothesis 2b.** *In crisis periods, the exploitative capability to efficiently manage a firm's expenses is positively associated with relative performance, such that organizations that can lower expense ratios are more likely to experience higher relative performance than organizations that cannot.*

### 2.3. Combined Contributions of Explorative and Exploitative Capabilities to Relative Sustained Performance

The flexibility to manage both revenue and expenses ambidextrously is a dynamic capability that can improve performance [16]. Questions arise about what performance contributions might be expected for an ambidextrous firm that attempts to simultaneously employ capabilities for both effectiveness and efficiency. A further question is whether the combined interaction effect is sub-additive, additive, or multiplicative (i.e., synergistic). There is ample theoretical evidence [26,27] to expect that all three outcomes are possible and/or plausible. If a firm increases its revenues and decreases its expenses, we expect the two capabilities to add up (assuming there are enough resources for investment and

development to work together). This additive performance outcome occurs because we are maximizing the difference (V − C) between the value created (V) and the cost (C) to produce the value [28]. Second, we might also expect a multiplicative interaction effect in that the combined increase in revenues and decrease in production costs results in increased income enhancing performance through synergies (assuming adequate resources exist for this purpose) [26,27].

Although we can assume that all firms have resource limitations, it is also possible to expect that many firms may lack the resources to simultaneously invest in effectiveness and efficiency capabilities adequately. Such inadequate simultaneous investment would logically result in a "stuck in the middle" position (e.g., [15]) where a firm incurs the capability costs but does not receive the benefits (in a competitive environment). Thus, the environment may have an adverse effect on the combined effect of the capability in an unpredictable manner. Investing in and deploying both capabilities deplete resources and could cancel out the effect of the other opposite strategy. Moreover, even if a firm possesses adequate resources to invest in both types of capabilities, doing so may produce a negative interaction effect. For example, in our earlier example in the previous section, reducing advertising expenses may help the bottom line right away, but it will also likely hurt the bottom line later by reducing revenues. This result could lead to a third possible outcome from the capability interaction. Conventional accounting logic would more likely assume a negative interaction effect over a positive interaction effect. However, a contrarian hypothesis would propose that the combined ambidextrous interaction effects of the dynamic capabilities to simultaneously manage a firm's revenue and expense ratios will be positively associated with relative performance. Hence, firms with higher levels of both capabilities should experience higher performance relative to their competitors than firms with lower levels of both capabilities. This logic suggests the third set of hypotheses.

**Hypothesis 3a.** *In stable periods, the combined contributions of the capabilities to (ambidextrously) manage both the revenue and expense ratios are positively associated with relative performance, such that organizations that can simultaneously increase revenue ratios and lower expense ratios will perform better than organizations that can manage one or the other but not both.*

**Hypothesis 3b.** *In crisis periods, the combined contributions of the capabilities to (ambidextrously) manage both the revenue and expense ratios are positively associated with relative performance, such that organizations that can simultaneously increase revenue ratios and lower expense ratios will perform better than organizations that manage one or the other but not both.*

### 3. Methods

We test these hypotheses using data from the credit union diversification project conducted by [29]. The original sample dataset included all financial reporting from the entire population of credit unions in the U.S. from 2000 to 2009. We selected a subsample of the largest credit unions (defined as those having assets over $100 million) to focus more directly on larger organizations that are more likely to have established routines and well-developed organizational capabilities. To examine our hypotheses, we compare two periods, one considered a period of economic stability from 2001 to 2004 and one considered a period of high economic instability, from 2006 to 2009 around the 2008 financial crisis and subsequent global economic recession. As the number of credit unions included in our sample varies by year, we only selected those credit unions with four years of complete data for each period (years 2001–2004 and years 2006–2009). Credit unions excluded because of missing information in one of the years represented less than 2% of the total number of credit unions. The final sample utilized for our study included 894 and 1127 credit unions in the first and second periods, respectively.

For both periods, we conducted hierarchical regression analysis to examine the individual and combined effects of the changes in the credit unions' revenue and expense ratios on their relative changes in ROA (RCROA) for the periods 2001 to 2004 and 2006 to 2009.

The hierarchical approach is particularly appropriate when analyzing potentially correlated independent variables [30]. Our analytical procedures consisted of five hierarchical models. Model 1 includes only the control variables. In models 2 and 3, we introduced the variables "Change in Total Income Ratio" and "Change in Total Expense Ratio," respectively. Model 4 includes both the income and expense ratio variables. In model 5, we added the ambidextrous interaction variable for the combined income and expense ratio variables to Model 4. In the model analysis, we used the variance inflation factors (VIFs) to assess multicollinearity. All scores were below 2, which is below the conventional VIF cutoff of 10.00, indicating that multicollinearity is not an issue [31].

### 3.1. Measures

### 3.1.1. Dependent Variable: Relative Change in Return on Assets (RCROA)

We operationalized relative performance as the relative change in ROA [32,33] of an individual credit union for the period (e.g., from the year 2006 to 2009 relative to the state average for the change in ROA from 2006 to 2009 (RCROA)). As credit unions are licensed to operate only within a particular state and not across state lines, the credit unions within a given state represent its direct competitors (whereas local, regional, and national banks and other alternative lending institutions represent indirect competitors). To calculate the relative change in ROA, we first calculated the absolute change in ROA for individual credit unions from 2006 to 2009. We then calculated the average change in ROA for all credit unions in a given state during the same period. We then subtracted the state average change in ROA from the individual credit union's change in ROA to determine the individual credit union's "performance" relative to its competitors in the state.

### 3.1.2. Independent Variables

There is no commonly accepted measure for dynamic capabilities in the strategic management literature; as such, scholars have operationalized dynamic capabilities in various ways [32,34]. Since dynamic capabilities are difficult to observe, it is common and acceptable for scholars to use proxies for the outcomes of such capabilities [35]. Therefore, in this study, we leverage such practices and utilize proxies for the outcomes of dynamic capabilities. To measure the underlying capability constructs, for explorative capabilities, we used the change in the total revenue ratio, and for exploitative capabilities, we used the change in the total expense ratio. Most of our measures have been previously used to assess the performance of financial institutions [36]. However, we are interested more in the change in these measures and their predictive effect on RCROA.

### Change in Revenue Ratio (CRR)

This measure is used as a proxy for explorative capability. We designed our measure for the change in revenues to measure the change in the total income to total assets ratio [37]. We calculated total income as net interest income (total interest received minus total interest paid) and non-interest income. We calculated the revenue ratio by dividing total income by total assets and then subtracted the 2006 revenue ratio from the 2009 revenue ratio to calculate the change in the revenue ratio for the crisis period. Similarly, we subtract the 2001 revenue ratio from the 2004 revenue ratio to calculate the change in the revenue ratio for the stable period.

### Change in Expense Ratio (CER)

This measure is used as a proxy for exploitative capability. We designed our measure for the change in expense ratio to measure the ratio of an individual credit union's total expenses relative to its total assets [38]. We calculated this ratio by dividing operating expenses plus interest expense by total assets. To determine the change in the expense ratio for the crisis period, we subtracted the 2006 expense ratio from the 2009 expense ratio. Similarly, we subtracted the 2001 expense ratio from the 2004 expense ratio to calculate the change in the expense ratio for the stable period.

Interaction of Explorative and Exploitative Capabilities

Ambidexterity (the interaction of the change in the revenue ratio (CRR) and the change in the expense ratio (CER)) is a term used to describe the use and balancing of two capabilities at the same time [16]. In this study, the two capabilities are explorative and exploitative capabilities, which are measured as the change in the revenue ratio and the change in the expense ratio, respectively. Using both capabilities involves an interaction effect. Therefore, our measure of ambidexterity for the interaction effect of the combined revenue and expense capabilities involves multiplying the change in the revenue ratio and the change in the expense ratio variables for both periods.

### 3.1.3. Control Variables

We controlled for a number of possible alternative effects. The first set of variables consists of credit union characteristics that might affect performance. Firm size might affect performance because different size credit unions may have different organizational structures. For example, large credit unions may benefit from economies of scale; at the same time, they might be hindered by a rigid organizational structure. In contrast, small credit unions may benefit from a more flexible structure and, as a result, be able to react more quickly to change [39]. In line with previous research, we measured firm size as the natural logarithm of total assets, after accounting for the change in assets observed for the period (e.g., the crisis period from 2006 to 2009). Therefore, we subtracted 2006 total assets from 2009 total assets to calculate the Change in Total Assets. Then, we calculated the natural logarithm of the change in total assets. Similarly, firm age might have an effect; therefore, we measure Firm Age as the number of years the credit union has been in existence until the beginning of the period (2001 or 2006).

Other possible effects from the external environment and not controllable by credit union management include environmental uncertainty and the geographical region in which the credit union is located. Because environmental uncertainty can affect performance [32,40], we control for Change in Uncertainty (U). To measure change in uncertainty, we subtracted the 2006 state unemployment rate from the 2009 rate. We also used regions (Northeast (NE), Southeast (SE), Midwest (MW), Southwest (SW), and Northwest (NW)) as dummy variables to control for the possibility of regional location differences. We present an overview of our variables and their measures in Table 1.

**Table 1.** Study Variable Descriptions and Measures.

| Dependent Variable | Description | Measures |
| --- | --- | --- |
| Relative Change in ROA (RCROA) | CU's (2006 to 2009) or (2001 to 2004) change in ROA relative to state average change in ROA (2006 to 2009) or (2001 to 2004). | e.g., [1] RCROA = Firm (2009 ROA − 2006 ROA) − State average (2009 ROA − 2006 ROA) |
| **Independent Variables** | **Description** | **Measures** |
| Change in Total Revenue Ratio (CRR) | Firm's 2006 to 2009 change in total income ratio (((interest received − interest paid) + non-interest income)/assets). | CIR = (2009 ((irec − ipaid) + noniy)/assets) − (2006 ((irec − ipaid) + noniy)/assets) |
| Change in Total Expense Ratio (CER) | 1 − Firm's 2006 to 2009 change in total expense ratio ((operating expense + interest expense)/assets). | CER = (1 − ((2009 opexpens + ipaid)/assets)) − (1 − ((2006 opexpens + ipaid))/assets) for Firm |
| Interaction of CRR and CER | Interaction effect of Change in Total Income and Change in Total Expense Ratios | CIR × CER |

**Table 1.** *Cont.*

| Control Variables | Description | Measures |
|---|---|---|
| Change in Assets (Size) | 2006 to 2009 change in total assets | CA = 2009 total assets − 2006 total assets |
| Firm Age (Age) | Firm's age in 2006 in years | Age = Current Year "2006" − Founding Year |
| Change in Uncertainty (CU) | Change in state unemployment rate (2006 to 2009) as a proxy for level of environmental uncertainty | CU = 2009 state unemployment rate − 2006 state unemployment rate |
| Region | Regional dummy variables | NE, SE, MW, SW, and NW |

[1] Note: we use the crisis period (2006–2009) to demonstrate, i.e., measures for the normal period are calculated in same way.

## 4. Results

A quick comparison of the performance of credit unions between the two periods shows higher variability in ROA in the crisis period (2006–2009) than in the stable period (2001–2004), which suggests we could observe different results for the two periods. A comparison of CROA for the two periods is shown in (Figures 1 and 2) below.

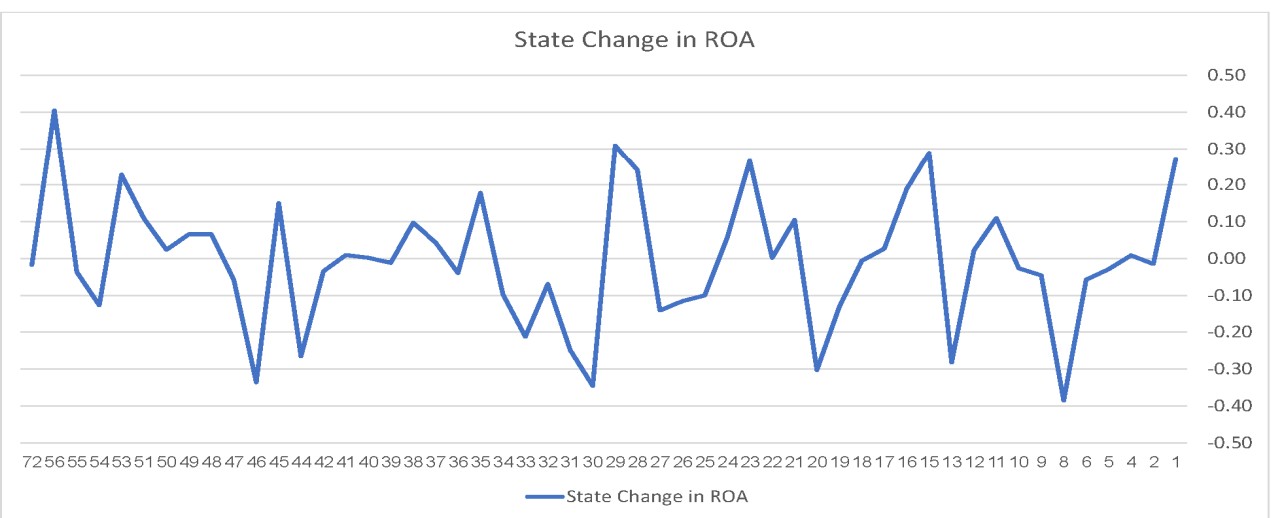

**Figure 1.** State CROA for the stable period.

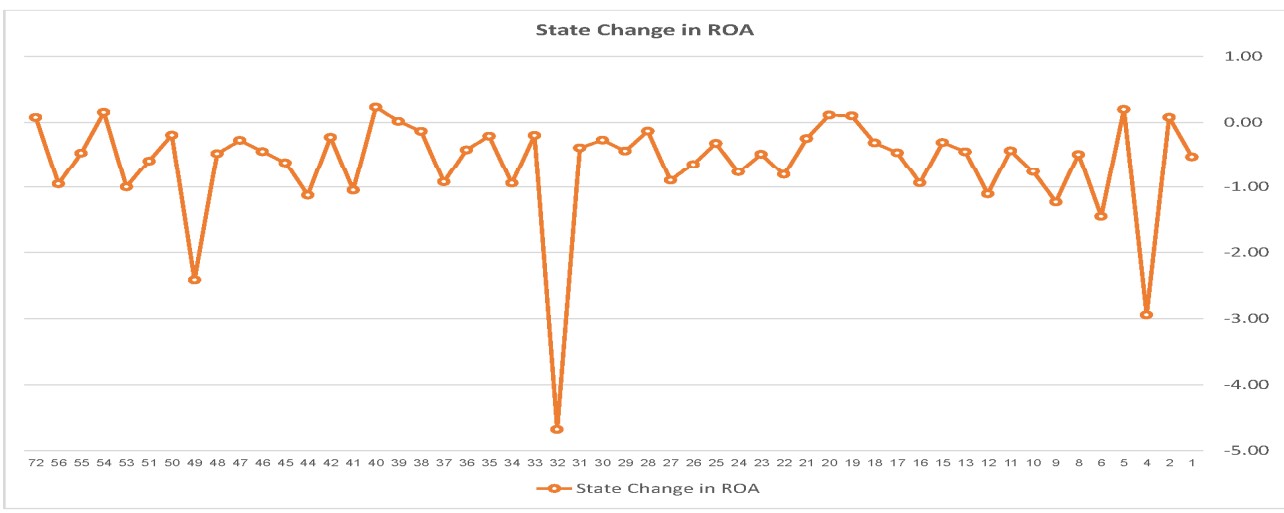

**Figure 2.** State CROA for the Economic Crisis period.

We report descriptive statistics and bivariate correlations among the study variables for the two periods in Tables 2 and 3, respectively.

**Table 2.** Means, standard deviations, and correlations (2001–2004) [a].

| # | Variable | Mean | SD | 1 | 2 | 3 | 4 | 5 |
|---|----------|------|-----|-----|-----|-----|-----|-----|
| 1 | Relative Change in ROA (RCROA) | 0.003 | 0.45 | | | | | |
| 2 | Change in Revenue Ratio (CRR) | 0.001 | 0.008 | 0.36 ** | | | | |
| 3 | Change in Expense Ratio (CER) | −0.00 | 0.006 | 0.05 | −0.76 ** | | | |
| 4 | Log (Change in Assets) | 8.515 | 0.34 | −0.09 ** | −0.09 ** | 0.07 * | | |
| 5 | Age | 56.30 | 13.70 | −0.00 | 0.06 | −0.05 | 0.04 | |
| 6 | Change in Uncertainty (CU) | 0.81 | 0.61 | 0.00 | 0.06 | −0.08 * | −0.07 * | 0.09 ** |

** $p < 0.01$. * $p < 0.05$. [a] Listwise, $n = 894$.

**Table 3.** Means, standard deviations, and correlations (2006–2009) [a].

| # | Variable | Mean | SD | 1 | 2 | 3 | 4 | 5 |
|---|----------|------|-----|-----|-----|-----|-----|-----|
| 1 | Relative Change in ROA (RCROA) | −0.001 | 1.24 | | | | | |
| 2 | Change in Revenue Ratio (CRR) | 0.001 | 0.016 | 0.08 ** | | | | |
| 3 | Change in Expense Ratio (CER) | 0.002 | 0.014 | 0.27 ** | −0.78 ** | | | |
| 4 | Log (Change in Assets) | 8.56 | 0.38 | 0.18 ** | 0.06 | 0.17 ** | | |
| 5 | Age | 60.96 | 13.70 | 0.01 | 0.01 | 0.05 | 0.07 * | |
| 6 | Change in Uncertainty (CU) | 4.62 | 1.84 | −0.08 ** | 0.08 ** | −0.07 * | 0.06 | −0.11 ** |

** $p < 0.01$. * $p < 0.05$. [a] Listwise, $n = 1130$.

In Table 4, we test Hypotheses 1a, 2a, and 3a for the period of economic stability. We first separately test the effects of revenue ratio and expense ratio on the dependent variable RCROA and then test the combined (interaction) effect on RCROA. The base model, Model 1, includes only the control variables and explains a very small amount of the variance in RCROA ($R^2 = 0.009$, $p > 0.01$). Model 2 includes the main effect of the change in the revenue ratio (CRR) on RCROA. Model 2 explains a significant amount of the variance in RCROA ($R^2 = 0.135$, $p < 0.01$). The added variance explained in Model 2 is over and above that explained by the base model ($\Delta R^2 = 0.126$, $p < 0.01$). In Hypothesis 1a, we proposed that the explorative capability to effectively manage revenues is positively related to relative performance in stable periods. The results reveal a significantly positive relationship between CRR and RCROA ($\beta = 0.36$; $p < 0.01$), supporting Hypothesis 1a. Model 3 includes the main effect of the change in the expense ratio (CER) on RCROA. Model 3 explains a very small amount of the additional variance in RCROA ($R^2 = 0.012$, $p > 0.01$), and the added variance explained is insignificant compared with that explained by the base model ($\Delta R^2 = 0.003$, $p > 0.01$). In Hypothesis 2a, we proposed that the exploitative capability to efficiently manage expenses is positively related to relative performance in stable periods. The results in Model 3 indicate a non-significant relationship between CER and RCROA ($\beta = 0.054$; $p > 0.01$), thereby rejecting Hypothesis 2a.

Model 5, the full model, includes the interaction between CRR and CER. In Hypothesis 3a, we proposed a positive interaction effect in that explorative capability to effectively manage revenues and exploitative capability to efficiently manage expenses combined would be positively related to relative performance in stable periods. We observe support for this hypothesis in the full model, as demonstrated by a significantly positive coefficient for the effect of the interaction term ambidexterity ($\beta = 0.057$; $p < 0.05$).

**Table 4.** Regressions Results for Normal Period Variables on Relative Change in ROA [1].

| Variables | Model 1 | Model 2 | Model 3 | Model 4 | Model 5 |
|---|---|---|---|---|---|
| Log (Change in Assets) | −0.126 ** | −0.063 * | −0.102 ** | −0.065 * | −0.071 ** |
| Firm Age | 0.000 | −0.020 | 0.009 | −0.020 | −0.020 |
| Change in Uncertainty | −0.001 | −0.030 | 0.003 | −0.014 | −0.015 |
| Northeast Region | −0.010 | 0.030 | −0.012 | 0.052 | 0.047 |
| Southeast Region | 0.017 | 0.012 | 0.014 | 0.003 | 0.006 |
| Midwest Region | −0.019 | 0.009 | −0.020 | 0.017 | 0.013 |
| Southwest Region | 0.004 | 0.003 | −0.004 | −0.017 | −0.019 |
| Change in Revenue Ratio (CRR) | | 0.360 ** | | 0.936 ** | 0.940 ** |
| Change in Total Expense Ratio (CER) | | | 0.054 | 0.759 ** | 0.768 ** |
| Interaction of CRR × CER | | | | | 0.057 * |
| $R^2$ | 0.009 | 0.135 ** | 0.012 | 0.378 ** | 0.381 ** |
| $\Delta R^2$ | 0.009 | 0.126 ** | 0.003 | 0.369 ** | 0.372 ** |

[1] Standardized regression coefficients are displayed; $n$ = 894. * $p < 0.05$. ** $p < 0.01$.

In Table 5, we test Hypotheses 1b, 2b, and 3b, for the period of economic instability. We first separately test the effects of revenue ratio and expense ratio on the dependent variable RCROA. We then test the combined (interaction) effect on RCROA. The base model, Model 1, includes only the control variables. Contrary to Model 1 in the stable period, Model 1 for the economic crisis period explains a significant amount of the variance in RCROA ($R^2$ = 0.037, $p < 0.01$), which could be the result of a significantly negative uncertainty variable (β = −0.093; $p < 0.01$). Model 2 includes the main effect of the change in the revenue ratio (CRR) on RCROA. Model 2 explains a very small amount of additional variance in RCROA ($R^2$ = 0.039, $p > 0.01$), and the added variance is insignificant compared with that explained by the base model ($\Delta R^2$ = 0.002, $p > 0.01$). In Hypothesis 1b, we proposed that the explorative capability to effectively manage revenues is positively related to relative performance in economic downturns periods. The results in Model 2 indicate an insignificant relationship between the CRR and RCROA (β = 0.054; $p > 0.01$), thereby rejecting Hypothesis 1b. Model 3 includes the main effect of the change in the expense ratio (CER) on RCROA. Model 3 explains a significant amount of the variance in RCROA ($R^2$ = 0.085, $p < 0.01$). The added variance explained in Model 3 is over and above that explained by the base model ($\Delta R^2$ = 0.048, $p < 0.01$). In Hypothesis 2b, we proposed that the exploitative capability to efficiently manage expenses is positively related to relative performance in crisis periods. The results reveal a significantly positive relationship between CER and RCROA (β = 0.22; $p < 0.01$), supporting Hypothesis 2b.

**Table 5.** Regressions Results for the Economic Crisis Period Variables on Relative Change in ROA [1].

| Variables | Model 1 | Model 2 | Model 3 | Model 4 | Model 5 |
|---|---|---|---|---|---|
| Log (Change in Assets) | −0.172 ** | 0.171 ** | 0.136 ** | −0.005 | −0.004 |
| Firm Age | −0.041 | −0.040 | −0.042 | −0.034 | −0.033 |
| Change in Uncertainty | −0.093 ** | −0.097 | −0.075 | −0.084 ** | −0.084 ** |
| Northeast Region | −0.004 | −0.005 | 0.011 | 0.044 | 0.042 |
| Southeast Region | 0.005 | 0.006 | 0.009 | 0.043 | 0.041 |
| Midwest Region | 0.008 | 0.008 | 0.011 | 0.011 | 0.010 |
| Southwest Region | 0.011 | 0.009 | 0.028 | 0.046 | 0.046 |
| Change in Revenue Ratio (CRR) | | 0.050 | | 0.845 ** | 0.844 ** |
| Change in Total Expense Ratio (CER) | | | 0.223 ** | 0.949 ** | 0.948 ** |
| Interaction of CRR × CER | | | | | −0.013 |
| $R^2$ | 0.037 ** | 0.039 ** | 0.085 ** | 0.283 ** | 0.283 ** |
| $\Delta R^2$ | 0.037 ** | 0.002 | 0.048 ** | 0.246 ** | 0.246 ** |

[1] Standardized regression coefficients are displayed; $n$ = 1127. ** $p < 0.01$.

Model 5, the full model, includes the interaction between CRR and CER. In Hypothesis 3a, we proposed a positive interaction effect in that explorative capability to effectively

manage revenues and exploitative capability to efficiently manage expenses combined would be positively related to relative performance in unstable periods. We do not observe support for this hypothesis in the full model, as demonstrated by an insignificant coefficient of the interaction term ambidexterity ($\beta = -0.013$; $p > 0.05$). Therefore, we reject Hypothesis 3b, which suggests that in economic crisis periods, using both capabilities together would not improve performance.

A summary of the results is shown in Table 6 below.

**Table 6.** Summary of the Results.

| # | Hypothesis | Decision |
|:---:|:---:|:---:|
| **1a** | *In normal periods, the explorative capability to effectively manage a firm's revenues is positively associated with relative performance.* | Accepted |
| **1b** | *In crisis periods, the explorative capability to effectively manage a firm's revenues is positively associated with relative performance.* | Rejected |
| **2a** | *In normal periods, the exploitative capability to efficiently manage a firm's expenses is positively associated with relative performance.* | Rejected |
| **2b** | *In crisis periods, the exploitative capability to effectively manage a firm's expenses is positively associated with relative performance.* | Accepted |
| **3a** | *In normal periods, the combined contributions of the capabilities (ambidextrously) to manage both revenue and expense ratios are positively associated with relative performance* | Accepted |
| **3b** | *In crisis periods, the combined contributions of the capabilities (ambidextrously) to manage both revenue and expense ratios positively associated with relative performance.* | Rejected |

## 5. Discussion

The results on their face value confirm previous studies in the literature showing that dynamic capabilities lead to better performance [9]. Nevertheless, we found that the types of capabilities that lead to better performance vary depending on the environment. The capabilities required in a stable environment are different from those needed in an economic downturn. Our study, therefore, contributes to the literature by suggesting different types of capabilities for different economic conditions. In addition, the results show that organizations should focus on either the explorative capability to increase revenues or the exploitative capability to decrease expenses, depending on their environment, which supports contingency theory, configuration theory, and the strategic fit arguments [9]. For the most part, we observe differences between the results from the periods of economic stability and instability. This observation is in congruence with the literature on strategic decision making; decisions are influenced by the environments in which decision making occurs [41]. The decisions regarding capability deployment to improve performance differed in the periods of economic stability and instability. Therefore, managers should always have a contingency plan for a crisis situation with a different set of decisions regarding capabilities and resources [42].

We observed that the variation in performance between credit unions is more volatile in the economic crisis period than in the stable period. Therefore, in general, credit union performance is more consistent in stable periods than in crisis periods. Second, in stable periods, the results show that the explorative capability to manage revenues is positively associated with performance. When conditions are stable, it is best for decision makers to find ways to increase revenues than to take a cost-cutting approach if seeking to improve relative performance. In the credit union context, the sources of revenue may include market expansion through increasing the member base or through alliances or simply through increasing interest income, non-interest income, and/or fee income; therefore, managers should encourage enhancing sources of revenue and growth. The results for the

stable period also show that exploitation and efficiency through cutting costs and lowering expenses do not make a significant contribution to performance.

However, the results for the economic crisis period are quite the opposite. In the crisis period, the exploitative capability to lower expenses is positively associated with performance. Additionally, in a crisis period, exploration through enhancing revenue does not make a significant contribution to performance. Therefore, in a crisis period, credit unions should focus solely on cutting costs (from sources such as interest expenses, non-interest expenses, and compensation expenses). In summary, in the credit union context, we observe that the organizational capabilities needed for each period are different. These results make logical, theoretical, and practical sense. In an economic crisis, resources are limited, and it is difficult to find new sources of revenue; thus, the best option is to exploit existing resources to efficiently manage/reduce expenses. These results contribute to the literature on dynamic capabilities and show that a different set of capabilities is needed for each period. Therefore, managers should be aware of the different sets of capabilities in their arsenal and be able to shift back and forth between these capabilities depending on the economic conditions of their environment [19].

Further, we also observed that the combined effects of strategic agility or ambidexterity to simultaneously employ both types of dynamic capabilities significantly enhances the relative performance contributions of the capabilities in stable environments. Using both types of capabilities (ambidexterity) is best when the environment is stable, which is shown by the significance of the interaction term in the stable period. However, we observe the opposite for the economic crisis period, as shown by the insignificance of the interaction term in that period. During an economic crisis period, higher performance can only be achieved by employing either the dynamic exploitative capability to decrease expenses but not, the dynamic explorative capability to increase revenue or both together, contradicting the ambidexterity arguments. This finding is interesting, but not surprising given the limited resources and high uncertainty during an economic crisis. While a firm should probably take action of some kind, conflicting (sub-additive) actions that theoretically use the same resources differently offer far less contribution than using one strategy over the other or combining the strategies. Therefore, rather than being "stuck in the middle" by trying to be agile and doing some of both, in a crisis, it is better to focus on one capability, particularly economizing [14].

It is interesting to note our finding that higher relative performance is achievable by simultaneously cutting costs and increasing revenues only in stable economic conditions. These findings challenge the classic position and support that exploiting two "opposite" strategic actions is a bad strategy (e.g., [15] arguments against being "stuck in the middle") in a stable environment. While the results support [15] arguments and continue to hold in environments with high economic instability, our results suggest that in stable environments, implementing both revenue enhancing and cost-cutting strategies simultaneously can improve firm performance more than implementing a singular revenue enhancement or cost reduction strategy. Finding strong evidence to support our ambidextrous or agility hypothesis is one of the major contributions of this research to the literature.

This study is one of the few attempts to merge the strategic management and crisis management literature in one study [43]. The results inform the literature on crisis management in situations such as the COVID-19 pandemic and the 2008 economic crisis regarding how decision makers should react to such crises [44]. We argue that, based on these results, decision makers should have a plan B that include a set of capabilities that differ from their plan A set of capabilities. This conclusion could be generalized to other industries and markets. The capabilities that work in stable periods differ from the capabilities that work in economic downturn situations, although the specific naming of these capabilities will vary from one context to another [18].

We believe we make a theoretical contribution to the literature on environmental dynamism as well as to the dynamic capabilities stream of research. Through this study, we show how firms could sustain performance using capabilities over a long period of

time since it is one of only a few empirical studies in these research streams that uses longitudinal data and a comparative approach between two periods of time.

*5.1. Practical Implications*

Our findings also have significant practical implications for the top management teams of credit unions. Businesses should have long-term plans that consider different scenarios and the different risks associated with each scenario to be prepared for fluctuations in the external environment, such as a global financial meltdown or the COVID-19 pandemic. Nair et al., 2014 [10] suggest utilizing enterprise risk management dynamic capabilities to cope better in an economic downturn. Questions about the kinds of managerial dynamic capabilities that will be suitable for such conditions may arise in difficult conditions (or bad scenarios) [42,45]. Management must question whether it continues with the same long-term strategic plan or should adjust the plan according to the new crisis condition.

In addition, as implied in the discussion, periods of high environmental dynamism provide important strategic opportunities for credit unions. As the results suggest, during a crisis period, credit unions tend to use a focus strategy where leaders move aggressively to respond to the situation. In fact, our findings clearly suggest the need for development of a strategic opportunity plan. Stated simply, as shown in Figure 3 below, credit union top management teams should prepare a specific version of the following plan to use when the appropriate need arises. While the specific actions followed need to be tailored to the dynamic capabilities of a specific credit union, the guiding focus of primarily on reducing costs over increasing revenues during a crisis.

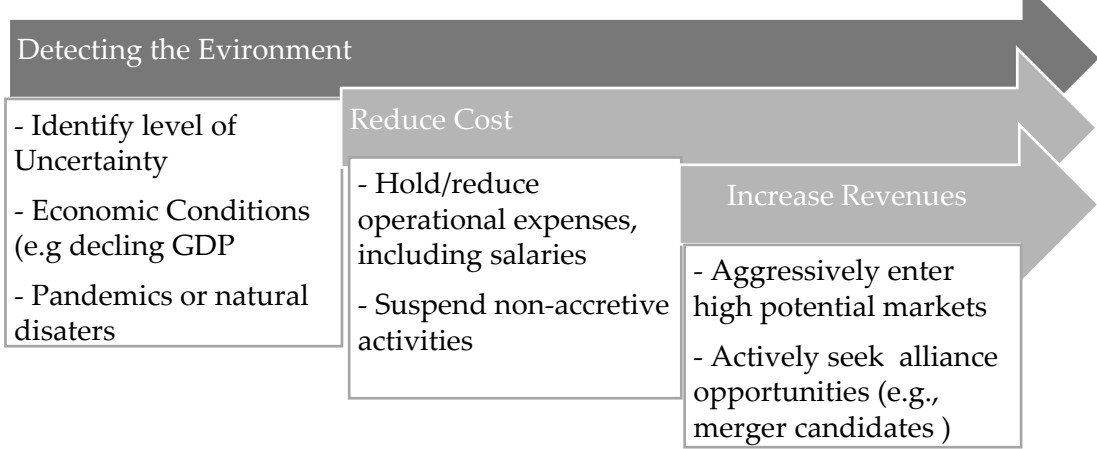

**Figure 3.** Practical Implications of Managing Dynamic Capabilities in an Economic Crisis.

Less dramatic, but equally practical, credit union leaders may want to develop crisis plans to implement in times of crisis. First, management should establish a unit to detect the environment and warn of any forthcoming crisis. Second, due to risk aversion during a crisis and limited resources, managers should activate the capability to reduce costs and expenses. They should dig deeper into operating expenses (e.g., salaries) and try to reduce expense ratios such as non-interest expense ratios. Last, when the crisis begins to recede, credit unions must find ways to produce revenues; this is possible by using revenue generating capabilities such as increasing the fee income ratio. It is interesting to speculate that this latter category, which includes late payment fees and charges for insufficient funds, may almost automatically become a significant revenue generator in stable times. Stated simply, payment of these fees by credit union members increases only in stable environments.

*5.2. Limitations and Future Research*

This study, like most, has some limitations. One limitation is that these credit unions are not limited to these types of dynamic capabilities. Other types of dynamic capabilities

could be studied. Given that there are various types of dynamic capabilities, and they are context-dependent, further research is needed to study different kinds of dynamic capabilities in different contexts.

Another limitation is that we use proxies to measure the outcome of the dynamic capabilities rather than directly measuring the capabilities themselves. Although our measures capture and improve our understanding of the outcomes of dynamic capabilities, it would be interesting if future research could find a means to directly measure the capabilities themselves that contribute to the outcomes (e.g., changes in revenues and expenses). Qualitative research is likely the best approach to investigate such unobservable resources and capabilities [46]. Further, our study included two periods of analysis, examining the changes in our sample from 2001 to 2004 (a stable period) and from 2006 to 2009 (before and after the 2008 recession). For future research, it would be interesting to study additional periods, such as before and after the COVID-19 pandemic to see if our results hold. In addition, because it is well established in the literature that the environment and the changes in the environment affect dynamic capabilities, it is recommended to use longitudinal data in future research to capture these environmental factors.

It would also be interesting for future research to examine the moderating effects of various types of uncertainty (e.g., macro, regional, industrial, technological, etc.) on the contributions of dynamic capabilities to performance. Finally, our sample included only credit unions, and while we believe that credit unions, to some extent, are representative of most organizations in the financial industry, we should be cautious when generalizing to other industries. As such, future research may consider extending our study to investigate dynamic capabilities and their interaction effects in other industries [34].

## 6. Conclusions

In this study, we examined the direct and combined contributions to performance by two of a firm's core capabilities: (1) the explorative capabilities to manage revenues (e.g., increase sales and improve cash inflows) and (2) the exploitative capabilities to manage expenses (e.g., reduce costs and improve efficiency, decreasing cash outflows). These were examined in two distinct periods: one stable and one in an economic downturn. While previous research has improved our understanding of the state of empirical support for such capabilities' contributions to performance [47], and the conditions and limits of these contributions [32], in this study, we theorized and examined both the individual and combined performance contributions of these capabilities in two dissimilar situations to improve performance through different economic conditions. By doing so, we improve chances of firms to survive and sustain performance in the long term. In addition, we validated that the need for one type of dynamic capability versus another may depend on the situation, that is, the individual contribution effects of capabilities, providing empirical support for the conclusions of recent theoretical reviews (e.g., [47,48]). We also offered empirical evidence that the ambidextrous interaction effects of these capability combinations are not additive or synergistic, but in fact, could be subtractive (offering support for [15,26,27]).

For many companies, dealing with economic crises (e.g., the 2008 recession, COVID-19 crisis, etc.) has been a challenge. Managers often continue trying to increase revenues for short-term gains instead of cutting costs and expenses, which could harm the company in the long run. Our study found support for the performance benefits of choosing and focusing on one set of capabilities (efficiently managing expenses) over another (effectively managing revenue), or trying to ambidextrously balance both capabilities, especially in a crisis (a highly uncertain environment), while ambidexterity appears to work well in economically stable situations.

Our findings also indicate that environmental recognition and the ability to act rapidly may be the most important dynamic capability that an institution holds in a time of crisis. Recognition is more important than denial; action is more important than inaction. Credit unions that acted quickly to strategically cut costs early in the financial crisis performed better than those that did not, or focused on trying to enhance revenues solely or in

combination with cost cutting. However, credit unions that sought to increase revenues, or do both by simultaneously cutting costs and increasing revenues performed best in stable environments. We hope that our work here provides further evidence supporting the usefulness of such dual capabilities to inspire and improve further research and practice.

**Author Contributions:** Conceptualization, P.D., W.J. and R.D.; Methodology, A.O.J.; Validation, A.O.J., P.D., W.J. and R.D.; Investigation, A.O.J., P.D., W.J. and R.D.; Resources, P.D.; Data curation, P.D.; Writing—original draft, A.O.J. and P.D.; Writing—review & editing, A.O.J., P.D., W.J. and R.D.; Supervision, P.D. All authors have read and agreed to the published version of the manuscript.

**Funding:** This research received no external funding.

**Institutional Review Board Statement:** N/A no human or animal involved, and we used secondary data.

**Informed Consent Statement:** No humans involved.

**Data Availability Statement:** The data is from the call reports for US credit unions at the National Credit Union Administration (NCUA). The NCUA call report data is publicly available at (www.ncua.org).

**Conflicts of Interest:** The authors declare no conflict of interest.

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
