# Peer review of "Examining the Sustainability of Contributions of Competing Core Organizational Capabilities in Response to Systemic Economic Crises"

_sustainability, doi:10.3390/su15054526_

Round 1

Reviewer 1 Report

The paper needs a thorough proof reading. For example in the first paragraph:

In the long run, organizations face a changing external environment. Economic conditions change every 10-year cycle, and pandemics and natural disasters occur more frequently. It is expected that firms will face increasingly more uncertainty in the future, as more crises such as the Covid-19 pandemic occur and global changes on the rise. According to contingency theory, Organization decisions and deployment of capabilities are contingent upon the external environment [1].

Economic conditions change constantly, not every ten years

Global changes on the rise?  What do you mean by this?

Why is Organization capitalized?

The introduction is conceptually confusing and lacks focus. I would suggest re-writing it from scratch to better focus on the issues at hand.

Reading on, you develop two hypotheses which are identical other than the words stable and crisis – so you are testing if the same outcome holds in different periods? This seems odd and is not something I have commonly seen. There would normally be some hypotheses variation such that exploratory suits crisis and exploitative suits stable, or something similar.

Hypothesis 1a. In stable periods, the explorative capability to effectively manage a firm’s revenues is positively associated with relative performance, such that organizations that can increase revenues are more likely to experience higher relative performance than organizations that cannot.

Hypothesis 1b. In crisis periods, the explorative capability to effectively manage a firm’s revenues is positively associated with relative performance, such that organizations that can increase revenues are more likely to experience higher relative performance than organizations that cannot.

The quantitative sections seem fine. The data is relevant and robust and the testing seems to be competently managed.

My main suggestion – rewrite the introduction and hypotheses development areas. These are really confusing and unclear, often rhetorical with old and outdated references.

Author Response

Response to review 2

Thanks for your valuable comments that will truly helped in improving the paper

Comment

Response

In the long run, organizations face a changing external environment. Economic conditions change every 10-year cycle, and pandemics and natural disasters occur more frequently. It is expected that firms will face increasingly more uncertainty in the future, as more crises such as the Covid-19 pandemic occur and global changes on the rise. According to contingency theory, Organization decisions and deployment of capabilities are contingent upon the external environment.

Thank you. The introduction was re-written and was proofread by native speaker.

Economic conditions change constantly, not every ten years

Thank you. The introduction was re-written and was proofread by native speaker.

Global changes on the rise?  What do you mean by this?

Thank you. The introduction was re-written and was proofread by native speaker.

Why is Organization capitalized?

Error is corrected

The introduction is conceptually confusing and lacks focus. I would suggest re-writing it from scratch to better focus on the issues at hand.

Thank you. The introduction was re-written and was proofread by native speaker.

The quantitative sections seem fine. The data is relevant and robust and the testing seems to be competently managed.

We appreciate your comment.

My main suggestion – rewrite the introduction and hypotheses development areas. These are really confusing and unclear, often rhetorical with old and outdated references.

Thank you. The introduction and theory developments sections were re-written and was proofread by native speaker.

Reviewer 2 Report

Thank you for the opportunity to read the paper.

I think that some minor changes can improve the paper.

- TITLE - Your study's title is too long. Consider changing it to shorter and eye-catching title.

- ABSTRACT - Please add a clear research objective in the abstract. As this should be the starting sentence for the abstract, you have to try to sharpen its focus.

- Cross check all references within text with your reference list and make sure that all references used in within text are listed in your reference list and remove any uncited reference from the reference list. You must also make sure that each reference in your reference list is accurate and complete in terms of authors’ names, title, volume number, issue number, pages, publisher etc.   

- Check the number of words of your document.

- Make sure that the flow of your article is improved. Academic journals do not prefer short paragraphs with one-three sentences or long paragraphs longer than half page.

- LIMITATIONS. This section (included in conclusions) does not enrich the paper. It repeats the discussion and does not reveal how analysing the paper’s limits can help other authors to add value to the field of the research.

I suggest that “limitations” should be enriched with a future research agenda.

Best of luck in your review.

Author Response

Response to Comments and Suggestions for Authors

Comments

Response

Thank you for the opportunity to read the paper.

Thank you for taking the time to review our paper and for the valuable comments.

I think that some minor changes can improve the paper.

- TITLE - Your study's title is too long. Consider changing it to shorter and eye-catching title.

Thank you, we changed the title to "Examining the sustainability of contributions of competing core organizational capabilities in response to systemic economic crisis”

- ABSTRACT - Please add a clear research objective in the abstract. As this should be the starting sentence for the abstract, you have to try to sharpen its focus.

We focused the beginning of the abstract with the dynamic capability prospective

- Cross check all references within text with your reference list and make sure that all references used in within text are listed in your reference list and remove any uncited reference from the reference list. You must also make sure that each reference in your reference list is accurate and complete in terms of authors’ names, title, volume number, issue number, pages, publisher etc.  

We updated it and checked all references

- Check the number of words of your document.

We checked the number of words it's around 8000

- Make sure that the flow of your article is improved. Academic journals do not prefer short paragraphs with one-three sentences or long paragraphs longer than half page.

we re-edited the whole article and rewrote introduction and theory section. Also, re-edited by a native speaker.

- LIMITATIONS. This section (included in conclusions) does not enrich the paper. It repeats the discussion and does not reveal how analysing the paper’s limits can help other authors to add value to the field of the research.

We highlighted some of the limitations and encouraged researches to repeat different context and overcome some of the limitations.

I suggest that “limitations” should be enriched with a future research agenda.

We added future agenda

Best of luck in your review.

Round 2

Reviewer 1 Report

Thank you for the changes which I think have improved the paper significantly.